# Effect of Fetoscopic Laser Photocoagulation on Fetal Growth and Placental Perfusion in Twin–Twin Transfusion Syndrome

**DOI:** 10.3390/jcm11154404

**Published:** 2022-07-28

**Authors:** Yao-Lung Chang, Chin-Chieh Hsu, An-Shine Chao, Shuenn-Dyh Chang, Po-Jen Cheng, Wen-Fang Li

**Affiliations:** Department of Obstetrics and Gynecology, Chang Gung Memorial Hospital, Linkou Medical Center, College of Medicine, Chang Gung University, 5, Fu-Shin Street, Kweishan, Taoyuan 333, Taiwan; b9602047@cgmh.org.tw (C.-C.H.); aschao1295@cgmh.org.tw (A.-S.C.); gene@cgmh.org.tw (S.-D.C.); pjcheng@cgmh.org.tw (P.-J.C.); wanna0507@gmail.com (W.-F.L.)

**Keywords:** twin–twin transfusion syndrome, fetoscopic laser photocoagulation, intrauterine growth restriction

## Abstract

Objective: To investigate the fetal growth pattern after fetoscopic laser photocoagulation (FLP) in twin-twin transfusion syndrome (TTTS) and the effect of FLP on placental perfusion and intrauterine growth restriction (IUGR) incidence. Methods: TTTS cases with a live delivery of both twins at least 28 days after FLP and with a neonatal follow-up at our hospital at least 60 days after delivery were included. The biometric data obtained before FLP (based on ultrasound); time point M1), upon birth (M2), and at neonatal follow-up (M3) were analyzed. The body weight discordance (BWD) was defined as (estimated fetal weight [body weight] of the recipient twin − estimated fetal weight [body weight] of the donor twin)/(estimated fetal weight [body weight] of the recipient twin) × 100%. Total weight percentile (TWP) was defined as the donor + recipient twin weight percentile; the TWP indirectly reflected the total placental perfusion. Results: the BWDs decreased from M1 to M2 to M3 (24.6, 15.9, and 5.1, respectively, *p* < 0.001, repeated measurements). The weight percentiles of recipient twins decreased after FLP, that is, from M1 to M2 (53.4% vs. 33.6%, respectively, *p* < 0.001, least significant difference [LSD] test). However, the weight percentiles of donor twins increased after delivery, that is, from M2 to M3 (13.2% vs. 26.2%, respectively, *p* < 0.001, LSD test). Moreover, the TWPs decreased after FLP, that is, from M1 to M2 (66.2% vs. 46.8%, respectively, *p* = 0.002, LSD test) and increased after delivery, that is, from M2 to M3 (46.8% vs. 63.2%, respectively, *p* = 0.024, LSD test). The IUGR incidences in donor twins were significantly lower after FLP (77.4% vs. 56.6%, respectively, *p* = 0.019, McNemar test) and further decreased after delivery (56.6% vs. 37.7%, respectively, *p* = 0.041, McNemar Test); however, no significant difference was observed in recipient twins’ IUGR incidences among M1, M2, and M3. The donor twin had catch- up growth in body weight, height, and head circumference after delivery, and the recipient twin had catch-up growth in only body height after delivery. Conclusions: the BWD decreased after FLP in fetuses with TTTS mainly because of the decreased weight percentiles of recipient twins. Moreover, it further decreased after delivery mainly because of the increased weight percentiles of donor twins. FLP not only decreased placental perfusion but also improved the TTTS prognosis because of reduced BWD and donor twin IUGR incidence.

## 1. Introduction

Twin–twin transfusion syndrome (TTTS) occurs in approximately 9% of monochorionic diamniotic twin pregnancies due to unbalanced intertwin placental blood flow [1]. Fetoscopic laser photocoagulation (FLP) is the first-line therapy for TTTS diagnosed before 26 weeks of gestation [2,3,4]. Compared with other treatments, such as amnioreduction, FLP results in a higher perinatal and both-infant survival rate [2] and a low neurologic morbidity rate of 4% to 18% [5]. However, studies on the prenatal and postnatal growth patterns of fetuses with TTTS treated using FLP remain limited [6,7,8,9]. FLP blocks unbalanced intertwin placental blood flow, but coagulation of the placental blood vessels can theoretically decrease the blood supply to either or both fetuses. Because fetal growth relies on placental perfusion, the growth of fetuses with TTTS before and after FLP may reflect the effect of FLP on placental perfusion. In this study, we used the donor twins’ weight percentiles and the recipient twins’ weight percentiles as the total weight percentile (TWP) to indirectly reflect placental perfusion and evaluate the effect of FLP on placental perfusion in fetuses with TTTS. 

Twin pregnancy is associated with higher perinatal mortality and morbidity than singleton gestation [10,11] because of the increased risks of preterm birth, intrauterine growth restriction (IUGR), and the unique complications of monochorionic gestations, such as TTTS and selective IUGR. IUGR is the primary cause of perinatal mortality and morbidity and affects approximately 7–15% of singleton and 25% of twin pregnancies [12,13]. Body weight discordance (BWD) is unique to twin pregnancies and may affect fetal and neonatal outcomes. Discordant growth in twin pregnancies is associated with fetal anomalies, IUGR, preterm birth, infection, and stillbirth [9,14]. Therefore, this study investigated the BWD and IUGR incidence among fetuses with TTTS before and after FLP.

To investigate the effect of FLP on TTTS, this study evaluated the biometric data, BWD, IUGR incidence, and TWP among fetuses with TTTS before FLP, after FLP, and at the neonatal follow-up. 

## 2. Materials and Methods

This retrospective study included fetuses with TTTS diagnosed before the gestational age of 26 weeks, who received FLP between October 2005 and May 2021, and with two survivals at least 28 days after surgery and neonatal follow-up data at least 60 days after delivery. The TTTS diagnosis was based on ultrasound findings as defined by Quintero et al. [15]. This study was approved by the Institutional Review Board (IRB) of the Chang Gung Medical Foundation (IRB number: 202200579B0). FLP was performed in the operating room under regional and local anesthesia by following the selective laser photocoagulation of communicating vessels method [16], with or without the use of the partial Solomon technique [17,18]; the FLP procedure for TTTS was described in a previous study [19].

The estimated fetal weight (EFW) was calculated using the formula outlined by Hadlock et al. [20], which combines the biparietal diameter, head and abdominal circumferences (HC and AC, respectively), and femur length. The fetal weight percentile was calculated using the World Health Organization’s fetal growth charts [21,22]. The postnatal growth percentile was calculated after adjusting for the neonatal weight as per the gestational age at delivery [23]. The neonatal body weight, HC, and body height (BH) were measured at our neonatal outpatient clinics. 

The EFW before FLP (measurement time point: M1); weight, BH, and HC upon birth (M2); and weight, BH, and HC at the neonatal follow-up (M3) were used to evaluate the fetal and neonatal growth patterns. If there was more than one set of data at M3, the maximum interval between the time points M2 and M3 was selected. 

The low Quintero stage was defined as Quintero stages I and II, and the high Quintero stage was defined as Quintero stages III and IV [24].

The BWD of a twin fetus was calculated as (EFW [body weight {BW}] of the recipient − EFW [BW] of donor twins)/EFW [BW] of the recipient twin) × 100% [25]. IUGR was defined as a weight percentile below 10%.

The BWD was analyzed according to the Quintero stage at the M1, M2, and M3 examinations in order to evaluate the effect of the Quintero stage on the BWD.

The TWP was calculated as donor + recipient twin weight percentile and was used to indirectly evaluate the placental perfusion function at the M1 and M2 time points.

Statistical analysis was conducted using SPSS (11.0 for Windows; SPSS Inc., Chicago, IL, USA). The data are expressed as mean ± standard deviation, median (min. and max.), and frequency (%), as and when appropriate. The qualitative data were compared using the χ^2^ test and Fisher’s exact test. The continuous variables were tested for normality. Repeated measurements were used to evaluate the weight percentiles, BWDs, and TWPs at M1, M2, and M3; the test was chosen on the basis of the sphericity assumption. If sphericity was not assumed, the Greenhouse–Geisser method was employed. After repeated measurements, a pairwise comparison of the two groups was performed using the least significant difference (LSD) test. Intergroup comparisons of the BWDs among the different Quintero stages were performed using a one-way analysis of variance (ANOVA), followed by post hoc least significance difference (LSD) tests for multiple comparisons. The homogeneity of the variances was tested. The McNemar test for related samples was used to investigate the IUGR incidences among donor twins at M1, M2, and M3. Student’s *t*-test and the Mann–Whitney U test were used to compare the continuous variables between the groups according to the normality of the data. We examined the normality of the data distribution using the Shapiro–Wilk test. A probability value of less than 0.05 was considered statistically significant.

## 3. Results

A total of 228 fetuses with TTTS who underwent FLP during the study period were included. After the exclusion criteria had been applied, 53 fetuses were included in the analysis (Figure 1). 

The basic characteristics of the included fetuses are listed in Table 1.

The BWDs and weight percentiles of donor and recipient twins at M1, M2, and M3 are listed in Figure 2. The BWDs at M1, M2, and M3 were 24.6, 15.9, and 5.1, respectively; a significant difference was discovered among the three measurements (*p* < 0.001, repeated measurements). The BWDs decreased significantly from M1 to M2 (M1 > M2, *p* = 0.004, LSD test), with a further significant decrease from M2 to M3 (M2 > M3, *p* < 0.001, LSD test). At M1, M2, and M3, the weight percentiles of the donor twins were 11.8%, 13.2%, and 26.2%, respectively (*p* < 0.001, repeated measurements), and those of the recipient twins were 54.4%, 33.6%, and 38.9% (*p* < 0.001, repeated measurements), respectively. The recipient twins’ weight percentiles were significantly lower after FLP (M1 > M2, *p* < 0.001, LSD test), with a nonsignificant difference found between M2 and M3 (*p* = 1.0, LSD test). The weight percentiles of the donor twins were not significantly different between M1 and M2 (*p* = 1.0, LSD test); however, a significant difference was discovered between M2 and M3 (*p* < 0.001, LSD test).

The TWPs at M1, M2, and M3 were 66.2%, 46.8%, and 63.2%, respectively, with the difference among the three measurements being significant (*p* = 0.02, repeated measurements, Figure 3). The TWPs decreased significantly from M1 to M2 (*p* = 0.002, LSD test) and then increased significantly from M2 to M3 (*p* = 0.024, LSD test). However, a significant difference was not found in the TWPs between M1 and M3 (*p* = 0.72, LSD test). 

With the analysis of the BWDs according to the Quintero stage, we found the BWDs were higher in Quintero stage III and IV cases at M1 and M2, but the BWDs were not significantly different among the Quintero stages after delivery. The results are listed in Table 2.

The HC and BH percentiles of donor and recipient fetuses at M2 and M3 are listed in Table 3. Significant increases in the HC and BH percentiles of donor twins were identified between M2 and M3; however, in recipient twins, a significant increase in only the BH percentile was discovered between M2 and M3.

The IUGR incidences among donor twins were 77.4%, 56.6%, and 37.7% at M1, M2, and M3, respectively; the incidences were significantly different among the three measurements (*p* < 0.001, chi-square test). The IUGR incidences among donor twins were significantly lower after FLP (M1 > M2, *p* = 0.019, McNemar test) and significantly decreased further between delivery and the neonatal follow-up (M2 > M3, *p* = 0.041, McNemar test). The IUGR incidences among recipient twins were 20.0%, 20.0%, and 18.7% at M1, M2, and M3, respectively; however, no significant difference in IUGR incidences was discovered among the three measurements (*p* = 0.96, chi-square test). 

## 4. Discussion

This study revealed the perinatal growth patterns of fetuses with TTTS after FLP; the BWDs of fetuses with TTTS gradually decreased from their pre-FLP value (24.6), to delivery (15.9), to the neonatal follow-up (5.1). Our results were similar to those of a previous study that reported the BWDs of fetuses with TTTS before FLP as 26.6 and those after FLP as 18.4 [7]. Moreover, the TWP of fetuses with TTTS were discovered to significantly decrease after FLP and then return to the pre-FLP level at the neonatal follow-up measurement. The IUGR incidences among donor twins were significantly lower after FLP, with a further significant decrease observed after delivery. However, the IUGR incidences among recipient twins were not significantly different between M1, M2, and M3. After delivery, the donor twins’ weight, HC, and BH percentiles were significantly higher than those before delivery; however, this difference was significant only for the BH percentiles of recipient twins. 

After FLP, the weight percentiles of recipient twins were significantly lower than before, but no significant changes in the weight percentiles of donor twins were observed. The decrease in the BWDs from M1 to M2 in our series was mainly due to the reduced weight percentiles of recipient twins. This may be attributed to the extra blood flow from the donor to the recipient twin through intertwin anastomoses before FLP; the extra flow ceased after FLP. After delivery, the weight percentiles of recipient twins did not significantly change, but the weight percentiles of donor twins were significantly higher; therefore, the decrease in BWDs from M2 to M3 was mainly due to the donor twins’ catch-up growth. The donor twin usually receives a smaller placenta share in the uterus than the recipient twin [10] but returns to the genetically determined weight after delivery. A 1986 study reported that the BWDs of 14 pairs of monozygotic twins were 28.3 at delivery, and they decreased to 9.6 at the age of 6 months [26]. The smaller twin in a twin pair tends to catch-up in growth after delivery. 

Theoretically, FLP may have negative effects on fetal growth because of the obliteration of communicating vessels in the placenta, leading to loss of associated placental territory and fetal growth impairment [7]. However, whether the placental perfusion decreases after FLP among fetuses with TTTS cannot be proven by analyzing the weight percentile change alone. We hypothesized that the TWP indicated the whole placental perfusion capacity in a twin pair and observed that the TWP significantly decreased after FLP. However, at the neonatal follow-up (M3), the TWP increased to its pre-FLP level (M1 level). This finding indirectly indicated that among fetuses with TTTS, FLP may lead to coagulated vessel loss associated with placental territory. The increase in the TWP at M3 may be attributed to the donor twin’s catch-up growth after delivery, because the weight percentile of the recipient twin did not significantly change after delivery. Therefore, recipient twins with TTTS usually have sufficient placental perfusion to achieve the genetically determined weight percentile in the uterus, but donor twins do not. 

The weight percentiles of donor twins were not significantly different between M1 and M2, but the IUGR incidences among donor twins decreased significantly from M1 to M2. This indicated that although placental perfusion decreased after FLP, the perfusion to the donor twin did not significantly decrease. According to the TWP analysis, although FLP decreased placental perfusion, it resulted in reduced BWDs and decreased IUGR incidences in donor twins. Therefore, FLP decreased the incidence of poor perinatal outcomes of TTTS [27,28]. 

The BWDs were found to be higher in Quintero stages III and IV of TTTS at M1 and M2. At M1, the BWDs may have been caused by imbalanced intertwin flow and placental territory discordance. Higher BWDs in Quintero stage III and IV cases may have reflected more severe imbalanced intertwin flow and placental territory discordance. After FLP, the placenta became functional-dichorionic, as each twin perfused its own amount of placenta [29]. At M2, the higher BWDs in Quintero stages III and IV of TTTS may have mainly reflected the placental territory discordance. At M3, both fetuses may have reached the genetically determined weights with significant donor catch-up growth; the BWDs were not significantly different among the different Quintero stage cases.

The reports about fetal growth in TTTS after FLP have been limited, and donor twins have been reported with catch-up growth after delivery [6,9]. In this study, we found the donor twins to have increased body weight, body height, and head circumference percentiles after delivery. We also found the recipient twins to not have significantly increased body weight and circumference percentiles, but to have significantly increased body height percentiles after delivery. The IUGR among donor twins with TTTS may have been because of a smaller placental territory and intertwin hemodynamic imbalance [30]. After FLP, the intertwin anastomoses became coagulated, so the intertwin hemodynamic imbalance was resolved, thereby reducing the IUGR incidence. However, the intertwin hemodynamic imbalance may not play a crucial role in the etiology of IUGR in recipient twins; therefore, the IUGR incidences did not significantly decrease after FLP. Studies reported that IUGR among donor twins is a risk factor for fetal demise after FLP [31] and 30 days after birth [32]. We only included two fetal survivals after FLP; therefore, the reported IUGR incidences among donor twins (less than the 10th percentile) with TTTS at M1 may have been underestimated in this study. 

This study had several strengths. First, the fetuses were followed up with at a single hospital, with the same measurement methods employed to evaluate the growth patterns. Second, we used the TWP to evaluate placental perfusion before and after FLP. The placenta perfusion in twin pregnancies is difficult to evaluate, but by using TWP, we proved that FLP may have the effect of eradicating placental perfusion associated with coagulated vessels. However, we also found that FLP may reduce the BWDs and IUGR incidences in donor twins, which could improve the perinatal outcomes in TTTS, despite losing placenta perfusion. Third, previous reports proved the donor catch-up growth after delivery [7,8], and we found that recipient twins had increased body height percentiles after delivery, which can be used for the consultation of the outcomes of TTTS patients before FLP. This study also had a few limitations. First, the EFW before FLP had certain limitations, though the intertwin BWD estimation through sonography is relatively accurate [33]. Second, we only included TTTS cases with two survivals after FLP; TTTS cases with a single survival after FLP may not exhibit the same growth patterns as reported in this study. Third, because of the limited data on twin pregnancies, especially on TTTS cases after FLP, we used the weight percentiles of singletons to evaluate the placental perfusion before and after FLP. The discordance of the weight percentiles between singleton and twin fetuses with TTTS after FLP cannot be excluded. Fourthly, this study lacked control evaluation of uncomplicated MC twins at M1, M2, and M3. A twin with TTTS before FLP was a monochorionic twin with intertwin anastomoses, and after FLP it became a functional-dichorionic twin. Whether the growth patterns of TTTS presented in this study were unique because of FLP treatment, or because they were similar to uncomplicated MC twins with similar BWDs as our TTTS cases at M1, cannot be answered by this study. Lastly, because of the limited number of fetuses with TTTS who underwent FLP and the strictness of the inclusion criteria, the sample in this study was small. 

## 5. Conclusions

After FLP, the BWDs of fetuses with TTTS decreased, which was mainly because of a decrease in the recipient twins’ weight percentiles. Moreover, the BWDs further decreased after delivery because of an increase in the donor twins’ weight percentiles. Although FLP may have decreased the placental perfusion in fetuses with TTTS, it may have improved the TTTS prognosis by reducing the BWDs and IUGR incidences in donor twins. After delivery, weight, BH, and HC percentiles increased in donor twins, indicating their catch-up in growth; however, only the BH percentiles increased among recipient twins. 

## Figures and Tables

**Figure 1 jcm-11-04404-f001:**
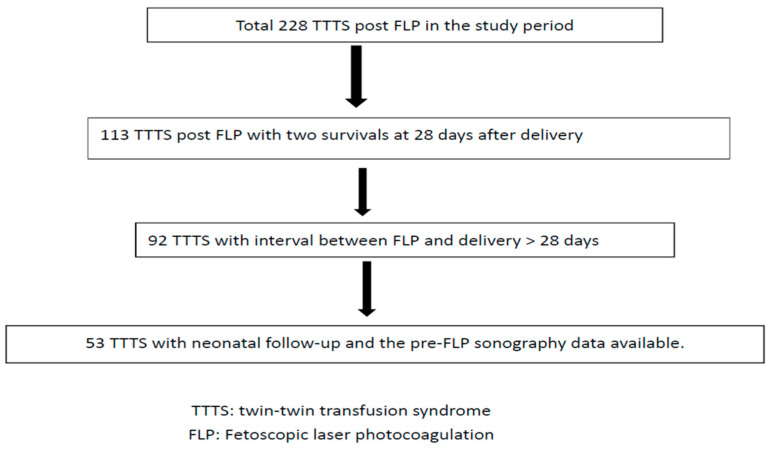
Flowchart of twin–twin transfusion syndrome post fetoscopic laser photocoagulation included in this study.

**Figure 2 jcm-11-04404-f002:**
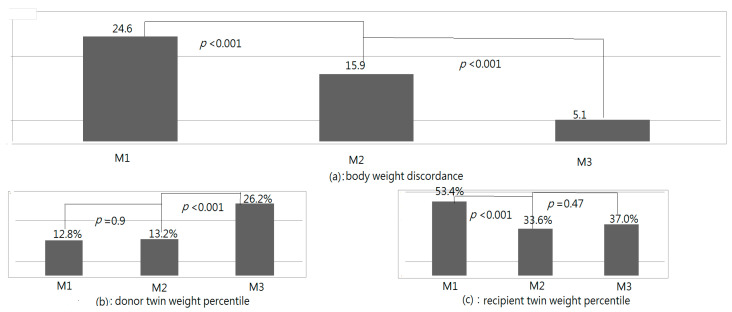
Body weight discordances (BWDs) and weight percentiles of donor and recipient fetuses (Table 1) upon birth (M2), and at neonatal follow-up (M3).

**Figure 3 jcm-11-04404-f003:**
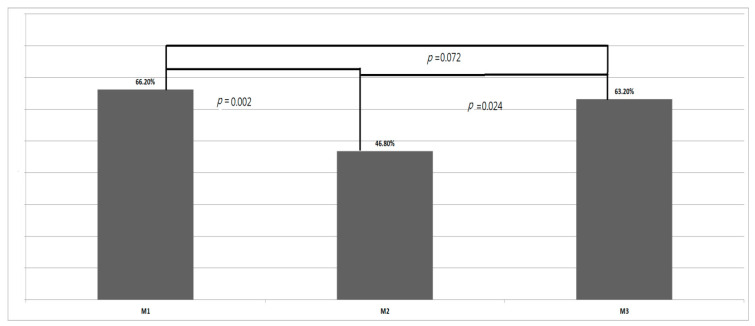
Total weight percentiles (TWPs) before fetoscopic laser photocoagulation (M1), upon birth (M2), and at neonatal follow-up (M3).

**Table 1 jcm-11-04404-t001:** Basic characteristics of fetuses with twin–twin transfusion syndrome after fetoscopic laser photocoagulation.

TTTS Receipted FLP with Two Survivals (N = 53)		
Maternal age at operation (year-old)		33.0 ± 5.1
Gestational age at operation (weeks)		20.6 ± 2.7
Gestational age at delivery (weeks)		33.5 ± 3.4
Interval from operation to delivery (days)		92.1 ± 48.7
Quintero stage (number)		
	I	8
	II	26
	III	13
	IV	6
Mean interval between delivery to neonatal follow-up (days)		569 ± 450

TTTS: twin–twin transfusion syndrome; FLP: fetoscopic laser photocoagulation.

**Table 2 jcm-11-04404-t002:** The BWDs at M1, M2, and M3 according to Quintero stage.

BWD	Quintero Stage I (N = 8)	Quintero Stage II (N = 26)	Quintero Stage III (N = 13)	Quintero Stage IV (N = 9)	*p*
M1	23.7 ± 12.6	18.8 ± 14.6	33.3 ± 12.8	32.0 ± 12.7	0.015
M2	10.7 ± 22.3	9.7 ± 17.2	25.7 ±15.0	28.7 ± 20.9	0.020
M3	2.9 ± 10.0	2.4 ± 10.5	11.4 ± 14.1	5.7 ± 13.1	0.157

Data are expressed as mean ± standard deviation. *p* values were generated by one-way analysis of variance (ANOVA). WDs at M1: Quintero stage IV and III > Quintero stage II (by least significance difference test). BWDs at M2: Quintero stage IV and III > Quintero stage II (by least significance difference test). The results of post hoc LSD tests for multiple comparisons can be found in the Appendix A.

**Table 3 jcm-11-04404-t003:** Head circumference percentiles and body height percentiles at M2 and M3.

	M2	M3	*p*
Donor head circumference percentile	26.3 ± 21.5	39.6 ± 30.1	0.004
Recipient head circumference percentile	47.1 ± 25.5	49.0 ± 31.0	0.66
Donor body height percentile	8.3 ± 12.2	33.6 ± 27.2	<0.001
Recipient body height percentile	18.6 ± 15.7	40.7 ± 24.6	<0.001

M2: upon delivery; M3: at neonatal follow-up; *p* values were generated using Student’s *t*-test.

## Data Availability

The datasets obtained and analyzed in this study are available from the corresponding authors upon reasonable request.

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
