# Peer review of "Effect of Fetoscopic Laser Photocoagulation on Fetal Growth and Placental Perfusion in Twin–Twin Transfusion Syndrome"

_jcm, 2022, doi:10.3390/jcm11154404_

Round 1
Reviewer 1 Report
This is a small retrospective study on the effect of laser coagulation on growth in TTTS pregnancies. Conclusions that were drawn in this study are more or less correct, but not surprising since other authors already investigated the effect of laser on placental territory.
First, many errors can be found in the text. Sentences are therefore hard to read. For example line 33-34, 44, 52-53, 201-203, 209-210, 213-214, 216-217 and the conclusion part. A native speaker could correct these errors.
In this study neonatal weight is included. However, how do the authors explain the influence of the placenta on the difference? Why is this parameter included? Neonatal weight is influenced by many different factors.
Can the authors elaborate on a possible difference between sFGR pregnancies and TTTS pregnancies, with regard to neonatal outcomes? Is there a difference?
Only 53 patients were included in this study. Only half of double survivors had neonatal follow up. I would suggest to include all patients with perinatal follow up (since the influence of the placenta is possibly measured during or directly after pregnancy), even though neonatal data are missing. Birth weight is a better estimate than weight at a few months for example.
Table 1: number of Q stages not corresponding to the number of included patients? 14 staged over 54 included patients?
Single survivors should be included, especially in this group there is a possible difference in placental territory and outcome.
Do the authors have data on incomplete lasers, the occurrence of post laser TAPS?
Author Response
First, many errors can be found in the text. Sentences are therefore hard to read. For example line 33-34, 44, 52-53, 201-203, 209-210, 213-214, 216-217 and the conclusion part. A native speaker could correct these errors.
Answer: Thanks for comment; we already have our manuscript edited by a native English speaker.
Regarding other comments' reply, please see the attachment.

Reviewer 2 Report
Dear Sir/Madame
I have read with a lot of interest your paper.
Your paper has in interesting enrolling a significant number of cases reported to the rareness pathology.
The study has the minus that is a retrospective study.It would be interesting to analyzed the results related to the Quintero stage , a significant parameter for the newborns prognosis.
The discussions and references needs to be updated – there are many recent articles in the field that came out in the last years.
Author Response
I have read with a lot of interest your paper.
Your paper has in interesting enrolling a significant number of cases reported to the rareness pathology.
The study has the minus that is a retrospective study.It would be interesting to analyzed the results related to the Quintero stage , a significant parameter for the newborns prognosis.
Answer: Thanks for the comment; we analysis the BWD according to high and low Quintero and we add the results in to table 2. And we add “Of the 53 cases included, 34 cases were low Quintero stage and 19 were high Quintero stage. The BWD was significantly higher in the high Quintero stage TTTS than in the low Quintero stage TTTS cases in M1, M2 but not in M3. And we also have added “Differentiation between TTTS Stages I vs II and III vs IV had been reported as not affecting the probability of double survival after laser therapy [24], and we also found the high Quintero stage was with low two survival rate in TTTS after FLP [29]. The BWDs were found as higher in the high Quintero stage TTTS than in the low Quintero stage TTTS in M1 and M2. In M1, the BWD may cause by imbalanced intertwin flow and placental territory discordance. Higher BWD in high Quintero stage cases may reflect more severe imbalanced intertwin flow and placental territory discordance. After FLP the placenta became functional dichorionic, as each twin perfused its own amount of placenta.[30] At M2, the higher BWD in the high Quintero TTTS may mainly reflect the more placental territory discordance. In M3, both fetuses may reach the genetically determinate weights with significant donor catch-up growth; the BWD was not significantly different between the high and low Quintero stage cases.” into the revised discussion section.
The discussions and references needs to be updated – there are many recent articles in the field that came out in the last years.
Answer: thanks for the comment; the reports about the pre-natal and postnatal growth patterns of fetuses with TTTS treated by FLP are limited. We try hardly to find the valid references about this manuscript, and we add new references and discussion into the revised manuscript. The reference number had been increased from 28 to 34.